**Subject Category:**
Biology (whole organism)

ecology/behaviour

spatial groups, communal denning, male associations, slender mongoose, *Galerella sanguinea*

**Author for correspondence:**
Beke Graw
e-mail: bekgraw@gmail.com

# Social organization of a solitary carnivore: spatial behaviour, interactions and relatedness in the slender mongoose

Beke Graw[1,2], Bart Kranstauber[1,2] and Marta B. Manser[1,2,3]

[1]Department of Evolutionary Biology and Environmental Studies, University of Zurich, Winterthurerstrasse 190, 8057 Zurich, Switzerland
[2]Kalahari Research Centre, Kuruman River Reserve, Van Zylsrus, Northern Cape, South Africa
[3]Mammal Research Institute, University of Pretoria, Pretoria, South Africa

BG, 0000-0001-9840-3243

The majority of carnivore species are described as solitary, but little is known about their social organization and interactions with conspecifics. We investigated the spatial organization and social interactions as well as relatedness of slender mongooses (*Galerella sanguinea*) living in the southern Kalahari. This is a little studied small carnivore previously described as solitary with anecdotal evidence for male associations. In our study population, mongooses arranged in spatial groups consisting of one to three males and up to four females. Male ranges, based on sleeping sites, were large and overlapping, encompassing the smaller and more exclusive female ranges. Spatial groups could be distinguished by their behaviour, communal denning and home range. Within spatial groups animals communally denned in up to 33% of nights, mainly during winter months, presumably to gain thermoregulatory benefits. Associations of related males gained reproductive benefits likely through increased territorial and female defence. Our study supports slender mongooses to be better described as solitary foragers living in a complex system of spatial groups with amicable social interactions between specific individuals. We suggest that the recognition of underlying 'hidden' complexities in these apparently 'solitary' organizations needs to be accounted for when investigating group living and social behaviour.

# 1. Introduction

Solitarily living species that have few social interactions except during mating and do not show cooperative behaviours [1] form the majority (80–95%) of carnivore species [2,3]. These species have received little attention in behavioural studies, even though it has been shown that some previously labelled as solitary have in fact more complex social organizations and structures, such as male alliance formation or spatial group formation [4–9]. Most likely 'group-living' and 'solitary' represent both ends of a multidimensional continuum [10,11] with solitariness relating to few direct interactions between conspecifics, while still allowing for an underlying complex social system [12,13].

According to Kappeler [14] social systems can be defined by four distinct components, the social organization, the social structure, the mating system and the care system. All of those describe the degree of social complexity a species exhibits [14] and social complexity is influenced by the number of individuals constituting a basic social unit and their interactions with each other [14]. To describe a species' social organization not only does the size of its social units need to be considered, but also its sexual composition and spatio-temporal cohesion [15]. Spatial organization in animals is believed to be closely linked to resource distribution. Home range size, shape and overlap are defined by resource availability and energy budgets [16–18], and are closely associated with a species mating system [19]. This frequently results in intersexual differences in spacing patterns. Females are usually directly affected in their reproductive output by the availability of ecological resources, such as food and shelter, while male reproductive output is limited by the access to receptive females [1,5,20–24].

If a limiting resource is predictable in space and time and restricted to a constrained area, this leads to territorial defence [25,26], while unpredictability or variation leads to overlapping ranges or roaming tactics [1,27]. When resources are patchily distributed in space and/or time the resource dispersion hypothesis (RDH) predicts that it is possible for one range to sustain more than the original owner [21,28], which can result in the formation of spatial groups [5,16,21]. Waser [29] extended this argument by pointing out that rapidly renewing resources might result in sharable ranges. Spatial groups are defined as animals with overlapping ranges but not necessarily temporal association and social interactions, that are spatially excluded from neighbouring spatial groups [21]. Solitary carnivores are expected to form spatial groups when the heterogeneity of resources is high and competition among residents for those resources is low [5,16,21].

Caro [30] suggested that the spatial patterns of female home ranges induce sociality in male carnivores. In solitary carnivores, male ranges are commonly larger than female ranges, overlapping those of several females [1]. As female densities increase, for example, in response to resource aggregation, a male's success at maintaining exclusive access to females may decrease, making it no longer possible to defend all females (especially in species with synchronized oestrus), or their home ranges against other males [31]. Such increased competition among males could result in the formation of male associations that formed to defend females or a large territory encompassing several females' home ranges against other males [21,32]. Examples for such male associations are found in cheetahs (*Acinonyx jubatus*) [33], the more gregarious lions (*Panthera leo*) [34], the Malagasy fosa [35] and have been suggested for several solitary mongoose species (Cape grey mongoose (*Galerella pulverulenta*) [36]; small Indian mongoose (*Herpestes javanicus*) [37]; black mongoose (*Galerella nigra*) [38]; slender mongoose [8,39,40]). In response to short-term aggregations of females, male raccoons (*Procyon lotor*) form temporary associations [5], while striped hyenas (*Hyaena hyaena*) [41] and slender mongooses [8] have been reported to form associations defined by spatial overlap, but few direct interactions, in human induced, rich resource patches.

The mongoose family (Herpestidae) consists of species covering the continuum from solitary, nocturnal, vertebrate feeders to highly social, diurnal insectivores [42,43], with the solitary lifestyle believed to be the ancestral one [44,45]. It has been proposed that anti-predator vigilance and defence is a main force for the establishment of groups in mongooses [2,3,45] and group living is associated with the use of open habitats [21]. Social mongooses like meerkats (*Suricata suricatta*), banded mongooses (*Mungos mungo*) and dwarf mongooses (*Helogale parvula*) have been the focus of extensive long-term studies [45–48], but substantially less is known about the less gregarious or solitary species [45,49].

Here, we investigate the socio-spatial organization of a small African mongoose, the slender mongoose, which has been described as mostly solitary living with potential male coalitions [8,40,42,50]. The slender mongoose is a pan-African species occurring throughout the continent south of the Sahara, with the possible exception of densely forested areas and extreme desert environments [50]. Little is known about its general biology and life history. Slender mongooses have been described as mostly diurnal, and as being more carnivorous than the more social mongoose species [39,50,51]. Its diet, the need to

stealthily stalk and hunt vertebrate prey, has been put forward as one hypothesis making solitary foraging and living necessary [43,52]. In a recent study, we have shown the slender mongooses in the Kalahari to be more opportunistic carnivores, also taking in a variety of invertebrates [53]. Previous studies on wild slender mongooses [8,39,40] were part of a larger study on mongoose species in the Serengeti, albeit largely based on anecdotal observations of individuals in close association to human settlements representing possibly rich food patches. These studies indicated possible male associations and cooperative behaviour, as well as social foraging in males [8,40], questioning the current description of this species as solitary and hinting at some degree of social flexibility.

We conducted the first long-term field study specifically targeting the slender mongoose, by investigating its spatial behaviour, frequency of social interactions and genetic relatedness in a population in the southern Kalahari, South Africa. To gain insight in the species social organization, we combined radio-tracking of collared individuals with a genetic analysis of relatedness to: (1) investigate home range size and overlap of slender mongooses, (2) characterize differences in ranging behaviour based on sex and seasonality, (3) study the stability of the spatial organization over time by investigating site fidelity, (4) identify patterns of relatedness within the population by looking at pairwise relatedness and (5) describe and characterize all observations of social interactions between mongooses.

# 2. Methods

## 2.1. Study population

We studied slender mongooses in the southern Kalahari in and around the Kuruman River Reserve (32 km$^2$; 26°58′ S, 21°49′ E), South Africa. The study area consists of a semi-arid landscape of sparsely vegetated dunes and herbaceous flats divided by a dry riverbed lined with scattered *Acacia* and *Boscia* trees (see details in [54,55]). There are two distinct seasons, a cold–dry winter from May to September and a hot–wet summer from October to April. The annual rainfall is typically around 250 mm and rain falls almost exclusively during the summer months [54]. During winter months, temperatures at night regularly drop below freezing. Slender mongooses in the Kalahari breed during the wet summer months (October–March). Females raise their offspring without the assistance of males. Pups are born in hollow trees in which they remain until emergence, followed by a period of foraging alongside mothers until independence around four months of age [53]. While all female offspring disperse as subadults, most males remain philopatric [56].

## 2.2. Capture procedure

Slender mongooses were captured for DNA sampling, individual marking and fitting of radio-collars throughout the study area in June–December 2007 and May 2008–May 2011. Traps were pre-baited with raw mincemeat, egg and small bones for three days prior to capture, followed by three days of capturing. Most captures were done in six focal areas, where adult individuals were fitted with radio-collars (14–18 g, Sitrack, Biotrack), and targeted specific animals using traps located at their sleeping sites or in locations that they often visited. To sample as much of the population as possible for genetic analysis, capture efforts were increased during winter months (May–September), when food availability was reduced and slender mongooses were more likely to enter traps. Winter captures also included areas not used by our main focal animals, longer capture sequences (5–7 days) and more traps/area (average distance between traps: 300 m). All captured mongooses were permanently marked with a subcutaneous microchip (Identipet®), sexed, measured and aged according to weight and tooth wear [53,56] before taking a small tissue sample (2–3 mm) of the tail tip. Samples were stored in 90% ethanol and frozen. We conducted a total of 215 captures, during which 131 individuals occupying an area of about 50 km$^2$ were sampled and identified. Based on recaptures and captures of new individuals, we estimate that we caught about 85–90% of all slender mongooses present in our main study area during the study period.

## 2.3. Spatio-temporal cohesion: range size and overlap

Whenever possible we radio-collared at least two adults that were caught in close proximity. We tracked a total of 34 adult slender mongooses (19 males and 15 females) between May 2008 and May 2011, using handheld two-element 'H' Yagi aluminium antennas (African Wildlife Tracking) connected to a

telemetry receiver (R-1000, Communication Specialists). Because it was not possible to habituate slender mongooses to a level where we could follow them and collect movement data during their foraging trips, tracking was done after mongooses had retreated in their sleeping burrows at night. From these sleeping locations, we calculated 95% kernel sleeping site ranges. We compared three different bandwidth estimators, $h_{LSCV}$, the reference bandwidth ($h_{ref}$) and $h_{plug-in}$, when analysing our data. Estimations with $h_{plug-in}$ resulted in highly fragmented ranges, while $h_{LSCV}$ failed to converge. Therefore, we had to use $h_{ref}$, even though this bandwidth estimator is known to sometimes result in over-smoothed ranges.

Ranges based on sleeping sites were calculated for all animals with at least 50 data points throughout our study and a minimum of 30 data points annually. Seaman et al. [57] proposed that the minimum number of data points needed to calculate kernel home range estimations is 30. We calculated annual ranges (2008–2009, 2009–2010, 2010–2011) and overall ranges based on the whole study period. Sleeping range overlap was determined for all animals alive at the same time, by calculating the area shared by two animals and then dividing it by the area of the total range of the focal animal. We used the 'kerneloverlap' function in 'adehabitatHR' [58] for this. We looked for patterns in range organization that would indicate spatial group formation by plotting ranges and comparing range overlap between individuals. Extensive overlap between certain individuals and exclusion/no overlap between others served here as our indicator for spatial groups.

To estimate site fidelity of animals, we compared range overlap between consecutive years. We divided the year into two seasons, breeding season from October to March and non-breeding season from April to September, to look at seasonal variation in sleeping range use. In a second step, we calculated changes in range overlap within and between spatial groups during the breeding and non-breeding season.

In addition to tracking slender mongoose in their sleeping sites, we collected data on foraging slender mongooses by triangulation. We established a system of two triangulation towers, consisting of two four-element Yagi aluminium antenna (African Wildlife Tracking) each. These were mounted parallel and out of phase to a 2 m pole that could be erected on sand dunes overlooking the area the targeted animal was expected in. Mounting the antennas out of phase allowed tracking with the 'null-method' where the signal of one antenna is subtracted from the signal of the second. When the direction of the target had been found, the signal from both antennas was exactly the same and the tracking signal disappeared. Listening for where the signal disappeared (null) instead of the loudest (peak) signal has been documented to be far more accurate [59]. Triangulation sessions lasted 1.5 h and each animal was located every 10 min. To ensure the animal did not move too far within one triangulation, bearings had to be taken within 2 min across receiver stations. Before data collection, we performed extensive tests with our system to minimize bearing errors and ensure we had the optimal locations for our towers. Locations of animals were calculated with LOAS 4.0.3.8 (Ecological Software Solutions LLC). Location estimations are most precise if bearings cross each other at an angle of 90° [59]; we only accepted locations based on bearing angles of 70°–135°. We triangulated animals between July 2010 and May 2011, targeting 16 animals in five spatial groups. During each session, we targeted all animals in a spatial group and neighbours they could overlap with.

## 2.4. Patterns of relatedness

We used microsatellite genotyping for our genetic analysis, using eight microsatellites found through cross-species amplification [56]. We investigated adult pairwise relatedness and relatedness patterns within eight different spatial groups on our study site and calculated reproductive success for all males. Pairwise relatedness coefficients using the Queller & Goodnight estimator [60] were calculated in GenAlEx 6.41 [61]. Parentage analysis was conducted with COLONY 2.0.1.3 [62,63] including all 131 individuals caught. We used the option to include known parents, as well as the exclusion of animals that are known not to be the parents. We only considered mongooses as potential parents of other adult mongooses if this matched our age estimations. To be assigned as parents, candidates had to have one or fewer mismatches with their potential offspring and a probability of more than 90% of being the parent. We used the following parameters to run COLONY: mating system: polygamy for males and females; medium likelihood precision; probability a parent is included in the candidates: males: 0.65, females: 0.75. All other parameters were left at the default settings. Error rates and allele frequencies are reported in Graw et al. [56].

## 2.5. Ethical note

Capture methods followed standardized procedures previously employed for the capture of yellow mongooses (Cynictis penicillata) [64] and were done by researchers (B.G.) after having received training

from experts at the Kalahari Meerkat Project. Slender mongooses were caught using box traps (Standard Humane Cage Trap, Animal Handling Support Systems, South Africa) covered in shade net to minimize the visibility of approaching researchers and reduce heat stress. Traps were checked every 20–60 min. Upon capture slender mongooses were immediately transferred into cloth bags to reduce handling time and simplify injections. Individuals were anaesthetized using an intra-muscular injection in the thigh of 2–6 mg kg$^{-1}$ Zoletil$^®$ (tiletamine–zolazepam, Virbac, Switzerland). Zoletil is a cyclohexamine drug that has been successfully used in a variety of carnivores. Induction time was on average two minutes. Morphological measurements, placing of microchips, DNA sampling, and collaring were all done while animals were fully anaesthetized. After procedures animals were placed in an aerated recovery box, closely monitored until full recovery and released at the capture site 45–90 min after capture. We observed no negative after-effects to the drug or invasive procedures, nor did animals become trap-shy. Tail clipping to gain DNA samples was chosen as non-invasive methods (through faeces collection or mouth swabs) were not feasible, and it is quicker and easier than collection of blood samples in these mongooses. During the procedure 2–3 mm of skin from the very tip of the tail were cut with a pair of sharp scissors, previously sterilized with an alcohol (90%) swap. Tail tips consisted of very dry, callus-like skin and clipping often caused no bleeding at all. In case bleeding occurred, it was easily stilled with a sterile gauze, afterwards the wound was sealed with skin glue.

Only adults were collared; females weighed at least 440 g while males weighed more than 570 g. Radio-collars, single-stage (Sirtrack) or two-stage (Biotrack) VHF transmitters, weighed between 14 and 18 g, equivalent to not more than 4% (less than the 5% recommended collar weight for mammals [65]) of an individual's body weight. Collared slender mongooses were observed to run, climb, carry pups and enter tree holes and bird nests without any obvious problems caused by the additional weight or shape of the collar. All collars were removed at the end of the study.

# 3. Results

## 3.1. Home range size and overlap

Over the whole study period of 3 years, we tracked the VHF collared slender mongooses 4392 times in their sleeping sites. Individual sleeping site ranges were calculated for 26 slender mongooses (14 males and 12 females; electronic supplementary material, table A) that were tracked at least 50 times during the study period (range: 51–412 trackings, mean $\pm$ s.e.: 126 trackings $\pm$ 25) and 30 times annually. Sleeping site home ranges for males were significantly larger than for females (mean annual ranges (mean $\pm$ s.e.): males $3.30 \pm 0.41$ km$^2$, females $1.22 \pm 0.01$ km$^2$, $p < 0.001$; whole study period: males $3.62 \pm 0.53$ km$^2$, females $1.18 \pm 0.15$ km$^2$, $p = 0.001$; electronic supplementary material, table A). Slender mongooses formed spatial groups consisting of one to three adult males that overlapped the ranges of several (1–4) adult females. Within our five main spatial groups (figure 1) female ranges were more exclusive, overlapping on average 12.3% annually, while male ranges showed a much larger overlap of 66.2% (table 1*a*). There was little overlap between individuals of neighbouring spatial groups with a mean annual of 5.6% (table 1*b*). Males shared about 11.2% of their range with males we considered to be outside their spatial group, while females only marginally overlapped (2%) with females in neighbouring spatial groups (table 1*b*; figure 1). This confirms our definition of spatial groups.

Range overlap between consecutive years was high, with males having an 83% overlap (three males in three consecutive years, one male in two consecutive years) and females an 85.5% overlap (one female in three consecutive years, and five females in two consecutive years; see also electronic supplementary material, table B). There was less overlap between core areas of consecutive years (males: 69.1%, females: 53.3%; electronic supplementary material, table B.2) but core areas remained within the total range of an animal (100% for males, 95.6% for females; electronic supplementary material, table B.3) from one year to the next. Including the sizes of 95% ranges in consecutive years, it seems that differences in overlap of ranges might be due to changes in range size rather than shifts of the areas an animal used. The alternative measure of estimating 'range drift' by calculating the shift of range centres between years showed for males an annual range centre shift of 164 m ($n = 7$, range: 27–314 m) and for females 231.57 m ($n = 7$, range: 40–503 m; electronic supplementary material, table C). Average range drift over the whole study period for these animals (comparing centroid shifts between the first and last year the animal was tracked) was 174.5 m for males ($n = 4$, range: 8–299 m) and 264.67 for females ($n = 5$, range: 65–503 m; electronic supplementary material, table C). These range drift distances suggest shifts in range use between years, but range centres did not move consistently and overall range drift remained

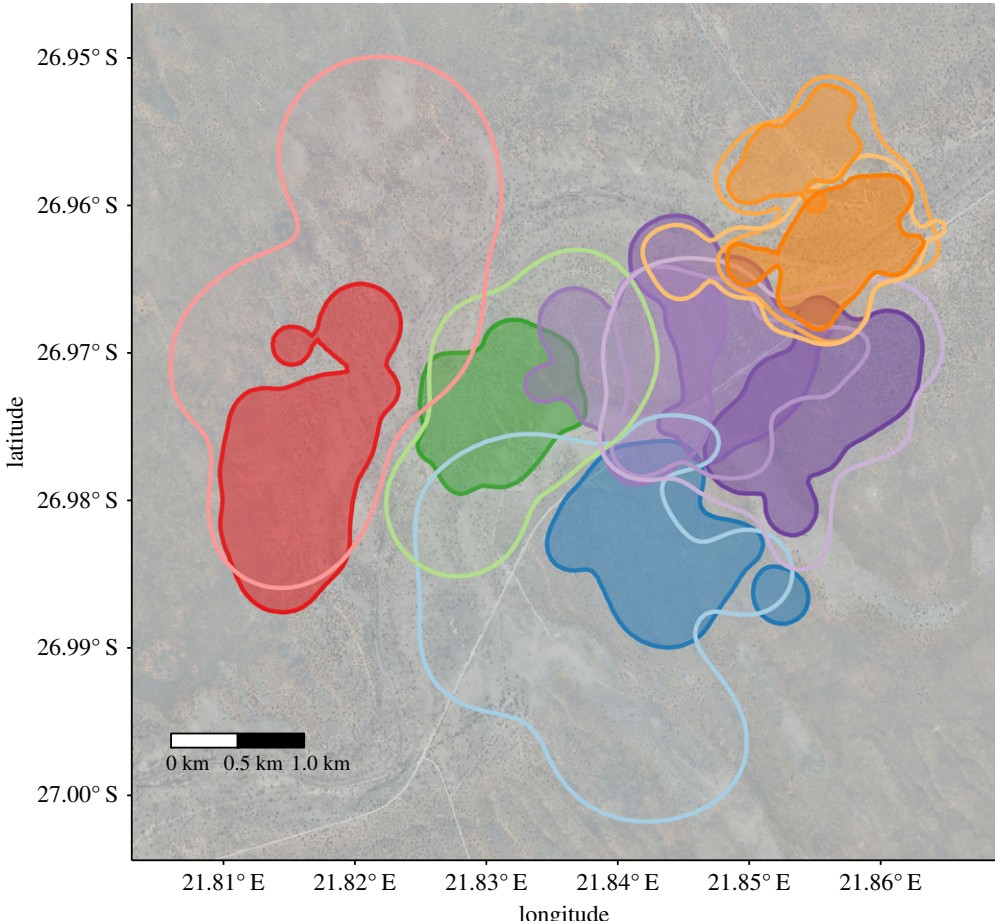

**Figure 1.** 95% sleeping ranges (whole study period using animals with at least 50 data points) for the animals originally caught in 2008 within the five main study groups. Empty ranges indicate males, female ranges are shaded, spatial groups are differentiated by colours (yellow, violet, green, blue and red).

**Table 1.** Average sleeping range overlap (based on at least 30 trackings annually) in % within (*a*) and between (*b*) spatial groups over the three different years, as well as the mean annual and total (over the whole study period) overlap. Male–male: overlap between male ranges, female–female: overlap between female ranges, male–female: male range overlapped by female range, female–male: female range overlapped by male range; sample sizes are given in parentheses.

|  | 2008–2009 | 2009–2010 | 2010–2011 | annual | total |
|---|---|---|---|---|---|
| (*a*) | | | | | |
| male–male | 67.8 (4) | 64.2 (6) | 66.7 (6) | 66.2 | 60.5 (8) |
| female–female | 17.9 (4) | 9.7 (8) | 9.2 (7) | 12.3 | 19.4 (10) |
| male–female | 39.4 (11) | 33.5 (13) | 32.3 (14) | 35.1 | 35.6 (20) |
| female–male | 79.6 (11) | 80.7 (13) | 76.4 (14) | 78.9 | 82.1 (20) |
| (*b*) | | | | | |
| male–male | 11.2 (27) | 12.7 (12) | 9.5 (18) | 11.1 | 6.2 (48) |
| female–female | 2.9 (20) | 2.1 (27) | 1.0 (26) | 2.0 | 1.2 (56) |
| male–female | 2.8 (24) | 4.8 (20) | 2.5 (24) | 3.4 | 2.3 (54) |
| female–male | 6.5 (24) | 7.3 (20) | 4.6 (24) | 6.1 | 4.2 (54) |
| overall | 5.8 | 6.7 | 4.4 | 5.6 | 3.5 |

low compared to range size (male ranges: 1–6.21 km$^2$; female ranges: 0.53–2.06 km$^2$) and the speed with which slender mongooses move (they are capable of moving several hundred metres within a few minutes; B.G. 2008–2010, personal observation).

**Table 2.** Range overlap (*a*) within and (*b*) between spatial groups as a function of season. Given are the annual average (annual) and total over the whole study period (total); sample sizes are given in parentheses.

| | annual | | total | | |
|---|---|---|---|---|---|
| | breeding | non-breeding | breeding | non-breeding | |
| *(a)* within | | | | | |
| m-m | 66.7 | 67.0 | m−m | 56.1 (8) | 62.7 (8) |
| f−f | 16.8 | 17.6 | f−f | 13.7 (10) | 21.6 (9) |
| m−f | 24.9 | 40.5 | m−f | 30.1 (19) | 33.4 (20) |
| f−m | 79.2 | 68.2 | f−m | 80.5 (19) | 71.9 (20) |
| *(b)* between | | | | | |
| m−m | 13.8 | 8.5 | m−m | 11.4 (34) | 9.7 (49) |
| f−f | 0.6 | 4.1 | f−f | 0.4 (55) | 2.5 (56) |
| m−f | 2.7 | 3.6 | m−f | 2.2 (47) | 2.7 (53) |
| f−m | 7.1 | 5.7 | f−m | 5.7 (47) | 7.2 (53) |

Male and female slender mongooses showed a reduction of around 50% in the number of different sleeping sites they used during the non-breeding season in comparison to the breeding season (seasonal average: breeding season: males 36, females 27; non-breeding season: males 18, females 14 burrows). Male ranges during the breeding season ($3.32 \pm 0.42\ km^2$ mean $\pm$ s.e.) were larger than during the non-breeding season ($2.70 \pm 0.67\ km^2$, $p = 0.030$). Female ranges did not change significantly in size (breeding: $1.04 \pm 0.11\ km^2$; non-breeding: $1.24 \pm 0.17\ km^2$, $p = 0.246$).

In line with bigger male ranges during the breeding season, their ranges overlapped more between spatial groups during the breeding season (annual average: breeding: 13.8%, non-breeding: 8.5%; total: breeding: 11.4%, non-breeding: 9.7%; table 2*b*). The increased male ranges during the breeding season and consecutive reduction during the non-breeding season were further reflected in a strong annual increase in overlap of male ranges by female ranges during the non-breeding season (breeding: 24.9%, non-breeding: 43.0%; table 2*a*). At the same time, the proportion of the range females shared with males was reduced (annual average: breeding: 79.2%, non-breeding: 68.2%; table 2*a*). Between spatial groups male range reduction was visible in a reduction in male–male overlap (annual average: breeding: 13.8%, non-breeding: 8.5%, total: breeding: 11.4%, non-breeding: 9.7%; table 2*b*). Female–female overlap between spatial groups did on the other hand seem to slightly increase during the non-breeding season (annual average: breeding: 0.6%, non-breeding: 4.1%, total: breeding: 0.4%, non-breeding: 2.5%; table 2*b*).

Within spatial groups slender mongooses shared sleeping burrows with up to five other mongooses on 16% of all nights they were located (range: 1–36%). Such communal denning was most common during winter months where animals shared dens on about 33% of all nights located (range: 0–77%). These values are minimum values as we could only detect communal denning if the denning partners were fitted with radio collars or we saw them emerge in the morning. A generalized linear mixed effect model with a logit link function including burrow ($n = 752$, fitted s.d. = 1.68) and animal ID ($n = 24$, fitted s.d. = 0.49) as random factors revealed a significant increase in communal denning with lower minimum outside temperature ($\exp(\beta_{intercept}) = 0.110$ (CI 0.063–0.175), $\exp(\beta_{temperate}) = 0.893$ (CI = 0.879–0.908), $p < 0.001$, $n = 4392$; figure 2). Results are similar when omitting opportunistic observations of den sharing. We detected sex differences in communal denning, while males shared burrows with other adult males (42%) and females (32%) of their spatial group, adult females only shared with adult males (27%) and independent juvenile offspring (44%) of both sexes, but never with other adult females. There were no significant differences in the number of communal denning events between the sexes (throughout year: males 16%, females 15% of nights located, $Z = -0.082$, $p = 0.936$).

Sample size for simultaneous triangulation of several animals was small ($n = 136$). In only 10 out of these 136 triangulation events (7%) were animals located within 100 m of each other, which we considered to be the maximum distance at which mongooses might be aware of each other and socializing, i.e. travelling together or foraging as a group. All these events were exclusively recorded during winter (month of July) and within one spatial group (M), though this might be due to the lack

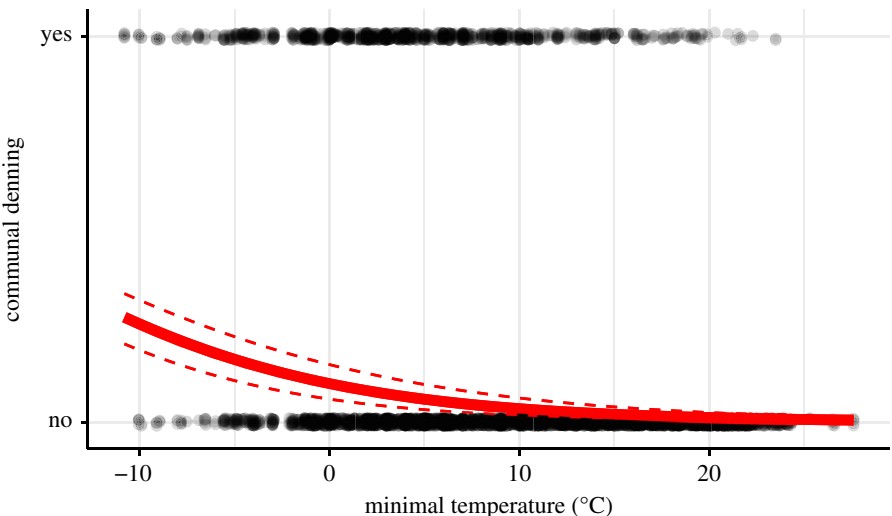

**Figure 2.** Communal denning in relation to outside minimum temperature (°C) for 24 different slender mongooses in six different spatial groups. The red lines show the effect of minimal nightly temperature and the confidence interval.

of winter triangulation data in other groups. It could indicate that socializing (social foraging) among adult slender mongooses is rare and limited to winter months, paralleling the increase of communal denning in winter. Eight of the socializing events happened between a male and a female (six were records from the same pair during the same session), two between males. On average animals were about 57 m apart when socializing (range 20–94 m).

## 3.2. Patterns of relatedness within spatial groups

Spatial groups consisted of one to three, often closely related adult males and one to four females whose ranges they overlapped. A Kruskal–Wallace $H$-test showed pairwise relatedness, based on 68 adult slender mongooses within eight spatial groups, differed significantly between the sexes ($\chi^2_{(2)} = 21.539$, $p < 0.001$). The average relatedness coefficient for male pairs within the same spatial group was $0.30 \pm 0.06$ (mean $\pm$ s.e., range: $-0.01$–$0.58$) and significantly higher than in the two other categories (*post hoc* Mann–Whitney $U$-test: male pairs versus female pairs: $Z = -4.64$, $p < 0.001$; male pairs versus male–female pairs: $Z = -2.52$, $p = 0.12$). Females on the other hand had a relationship coefficient of $-0.01 \pm 0.02$, showing no relatedness to each other. Male–female relatedness differed greatly between pairs and included closely related mother–son pairs alongside non-related mating pairs within one spatial group. On average, males and females had a relationship coefficient of $0.12 \pm 0.03$. This was significantly different from the average pairwise relatedness between female pairs (Mann–Whitney $U$ *post hoc*: $Z = -2.63$, $p = 0.009$). Parentage analysis revealed that the high male relatedness was explained by father–son pairs or (half) brothers sharing a spatial group range (figure 3).

All adult females within spatial groups bred, whereas males shared reproduction in some cases, but not in all. Of 33 adult males included in our paternity analysis, 16 (48%) were reproductively successful, i.e. we could assign them at least one immature individual caught during our study period. Number of sired pups per male varied from one to seven, with a median of one pup sired per adult male. This is based on 36 successfully assigned immature slender mongooses, an assignment success of only 55% which might be due to our high assignment criteria (probability of at least 90%, less than two mismatches) or not all males having been sampled. Based on our eight main groups, males that formed associations with close kin (spatial overlap, increased communal denning) seemed to have more females within their spatial group (2.75 versus 1.75) than males in spatial groups without close kin associates. Associations fathered on average more pups (5.75) than males without close kin (2.25). Looking at individual male success and including all males analysed in our paternity analysis, the average number of pups for an associated male was 2.5 (25 pups, 10 associated males), while this was only 0.48 for an unassociated male (11 pups, 23 males; table 3). Evidence is only anecdotal as sample sizes were small, but associated males possibly had a higher reproductive success rate (20% of associated males did not father a pup, while 65% of unassociated males did not father a pup; table 3).

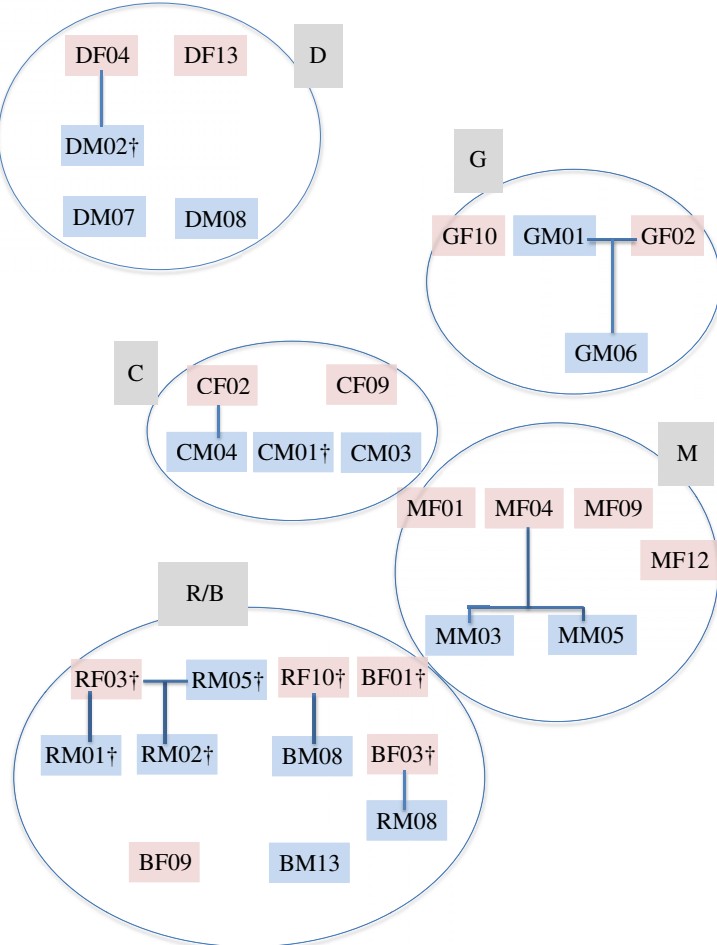

**Figure 3.** Relatedness based on parentage analysis within main study groups 2007 – 2011, for which also spatial data are reported. (Parent – offspring pairs are indicated, † denotes individuals that died/disappeared during the study, males are blue, females in red letters, arrangement of the circles around the individuals mirrors spatial organization of spatial group (D, G, C, M, R/B) ranges in the field, within circles spatial group members are arranged in rows according to time scale.)

**Table 3.** Comparison of individual reproductive success for all slender mongoose males analysed throughout the study based on association status.

|  | associated males | non-associated males |
|---|---|---|
| no. males | 10 | 23 |
| pups born to | 25 | 11 |
| average no. of pups/male (range) | 2.5 (0 – 7) | 0.48 (0 – 2) |
| no. of males with no pups | 2 | 15 |
| % of males with no pups | 20 | 65 |

Associated males also seemed to have a slightly higher success rate at preventing outside males from fathering pups within their spatial group (on average 15% of pups fathered by outside males versus 21% of pups fathered by outside males). In two groups where a father–son pair was present, the son did not reproduce within the spatial group (SGM06 in the G group, SRM02 in group R/B; figure 3). Mothers did not breed with their philopatric male offspring ($n = 3$ females, SDF04, SGF02, SMF04; figure 3) and mated instead with neighbouring males when no non-kin male was available (SMF04 in the M group; figure 3). We further have anecdotal evidence for multiple paternities within the same litter, where two littermates had different fathers. Out of four litters that had more than one pup and were completely sampled for DNA, two litters had multiple fathers (total number of emerged litters was 15), and therefore promiscuous mating of females.

# 4. Discussion

Slender mongooses in the Kalahari foraged solitarily and lived in a system of stable spatial groups where one to three adult males occupied large ranges that overlapped the exclusive ranges of several females. Besides being spatially separated from other spatial groups, these groups can be distinguished by their communal denning behaviour. Slender mongooses shared dens mostly during cold winter nights to potentially gain thermoregulatory benefits, and this only happened within a spatial group and never between groups. Within spatial groups, only males shared burrows intra-sexually, while adult females shared with adult males and independent offspring, but never with other adult females. Genetic analyses revealed a high relatedness of males within spatial groups, which in some cases led to male associations and, based on anecdotal data, possibly reproductive benefits, while females within a spatial group were unrelated.

## 4.1. Spatial organization

The spatial organization of slender mongooses at our study site seems to resemble the typical pattern of intra-sexual territoriality described in a variety of solitary carnivores, such as mustelid species, where females defend small, exclusive ranges, while males have larger, less exclusive ranges overlapping those of several females [1,66]. Ranges were stable over several years, with high site fidelity and shifts in range overlap between consecutive years mostly due to changes in range size, rather than the animals using a different area. This pattern most likely reflects sexual differences in resource competition. While females are limited in their reproductive output by ecological resources, such as food and shelter, males are limited by the access to reproductive females [67,68]. Male slender mongooses had significantly larger ranges during the breeding season than during the non-breeding season, indicating that they enlarge their range in response to the presence of receptive females. During the non-breeding season, food is most likely the main limiting resource, and ranges are smaller meeting metabolic demands [1,69]. By contrast, female slender mongooses showed no significant change in range size between breeding and non-breeding season, further supporting the hypothesis that females are distributed according to limiting ecological resources such as food [67,68]. Female slender mongooses rear pups on their own [53], and therefore their reproductive success is directly linked to the amount of energy they can allocate to reproduction and they should follow a behavioural tactic that maximizes the chances of securing enough food [1]. Patchily distributed food resources and/or rapidly renewable food sources, as in the habitat of slender mongooses, may allow several individuals to share the same range, according to RDH [21,28]. Original owners need to ensure their range encompasses enough exploitable food patches to sustain them at all times, which can lead to a surplus of resource over some or all of the time [1,21,28]. We found such range overlap leading to spatial group formation in the slender mongoose in the Kalahari.

## 4.2. Communal denning

Communal denning behaviour further supported our definition of spatial groups, as neither males nor females were ever observed to socialize and den with animals outside their spatial group. Communal denning in the Kalahari correlated with minimum temperatures, suggesting that mongooses gain thermoregulatory benefits from it. Socializing to gain thermoregulatory benefits has been described under the social thermoregulation hypothesis [70] and consecutively been found to explain communal denning in species such as Abert's squirrels (*Sciurus aberti*) [71]. Sharing a nest site also increases nest temperatures in southern flying squirrels (*Glaucomys volans*) [72]. As night temperatures in the southern Kalahari can drop well below freezing [54,55], we can assume significant energetic benefits from huddling together. This behaviour also coincides with longer sunning time in the morning before leaving to forage [53] and the use of only about half the number of different sleeping sites as opposed to summer months. Further studies are needed to investigate qualitative differences in the insulation of winter versus summer sleeping sites but Abert's squirrels do prefer better thermally protected cavity nests over dreys (twig nests) as ambient temperatures decrease [71].

## 4.3. Male associations

Our data suggest that males that shared a range and were closely associated with a male relative (father–son pairs, (half) brothers) showed reproductive benefits in terms of number of females in their spatial

group, number of matings lost to extra-group males (pups sired by outsiders) as well as the number of pups sired. While sample sizes were small for our study, it reflects what has been found in other carnivores showing male alliance behaviour. It also confirms data from previous studies on wild slender mongooses in the Serengeti [8,40]. Here, male associations were described, with ranges of up to four males overlapping 80–98% and associations being stable over up to 7 years [8]. Males behaved amicably and were observed to den, play and travel together [40]. We also have support for the existence of male associations that go beyond sharing a common range in our study. Anecdotal observations of socializing behaviour outside the shared dens included allogrooming and sunning together in males (B.G. 2018, unpublished data), and animals foraging in close vicinity of each other (assumed social foraging) from our triangulation data. Male alliances that benefit through cooperative territory and female defence, bigger territories including more receptive females, and increased offspring production have been described for cheetahs (*Acinonyx jubatus*) [32], lions (*Panthera leo*) [73] who also benefit through longer tenure as residents, raccoons [5] and fosas (*Cryptoprocta ferox*) [35,74]. As associated males in slender mongooses were closely related we can at least assume indirect reproductive benefits regardless of whether they gain direct matings. This has also been suggested for the striped hyaena, where several males in a coalition compete over only one female [41].

## 4.4. Female resource competition

Adult females seemed to avoid each other by having almost exclusive ranges and did not socialize. This might indicate a high intra-sexual competition over crucial resources. Besides food resources, females might be limited in the number of den sites to give birth in. Slender mongoose pups are born in hollow trees and remain in the tree while the mother goes off foraging until they are around 49 days old [53]. Hollow trees protect vulnerable pups from predators while the mother is away and suitable trees are presumably a limited resource. During the reproductive season, females are further limited in the distance they can travel during the day to forage, as they need to return to their pups at regular intervals to nurse them [53]. A female's range therefore needs to include enough food patches to support her and her pups throughout the year, all within a distance manageable while she needs to return to her stationary pups, and include trees suitable for pups to be born in. In contrast to the intra-sexual avoidance, females were tolerant towards males with whom they overlapped and associated with, within and outside the breeding season. While communal denning only occurred with males of their own spatial group, the genetics of the offspring revealed that females occasionally mated with males outside their own spatial group.

## 4.5. Slender mongoose social organization

We have shown that the social organization of slender mongooses in the Kalahari can best be described as a system of solitary foragers living in spatial groups. While females are intra-sexually intolerant and territorial, males can form associations with close kin due to philopatry [56] and presumably gain reproductive benefits through increased pup production and better defence of females against outsiders. Within spatial groups, animals socialize by denning communally, most likely as a form of thermoregulation. This happens inter-sexually, as well as intra-sexually in males. Females further den communally with independent offspring, including not yet dispersed female offspring. The slender mongoose social organization therefore shows some degree of flexibility in response to ecological variables such as outside temperatures (in males and females), as well as the presence of close kin (in males) leading to male philopatry and male associations. Strictly adhering to the definition of 'solitary' animals as never cooperating or socializing except during mating [1], slender mongooses are not solitary and it might be more appropriate to use the term 'solitary foragers'. Similar flexibility in social organization has been found in another solitary mongoose, the white-tailed mongoose (*Ichneumia alhicauda*), where females in high-density populations have overlapping ranges and form clans possibly through female natal philopatry [75]. Females belonging to the same clan tolerate each other, even occasionally share dens and behave aggressively towards females of other clans [75]. This strikingly similar (but in the reverse sex) pattern in the white-tailed mongoose has been explained by diet of rapidly renewing invertebrates allowing natal philopatry of females.

While the basic social unit [14] of the slender mongoose is one, and the quantity of social interactions might be lower than in more social mongooses, it still shows a surprising complexity in terms of the quality of interactions. Slender mongooses clearly distinguish between members of their own spatial group and outsiders, visible through the communal denning behaviour and spatial arrangement.

Genetic analyses revealed kinship structures and male philopatry [55] that would have remained hidden otherwise and males could associate with other males most likely to gain reproductive benefits. The slender mongoose provides a good example for the importance of taking the whole of a species social system (social organization, social structure, mating and care system [14] into account, as otherwise these more 'hidden' complexities might be missed [76].

## 4.6. Outlook

One shortcoming of our study might be the fact that we managed to gather only a limited movement dataset while mongooses were active during the day. Our range estimations based on the use of sleeping sites likely underestimate the true extent of space use by individuals. Nevertheless, we believe our sleeping site ranges did not differ greatly from home range sizes, as all observations of mongooses during the day were within calculated ranges, and the same was true for capture sites. Further studies using either habituated animals (though this will be very difficult; B.G. 2008–2010, personal observation), a more successful triangulation method for this habitat or using small GPS collars on slender mongooses might show ranges to be slightly bigger than in our study. This might lead to somewhat different but qualitatively similar results in range overlap; for example, more overlap in foraging ranges than sleeping ranges of neighbouring spatial groups is plausible.

# 5. Conclusion

Our results on the slender mongoose in the Kalahari fall in line with other studies on species formerly considered to be strictly solitary, such as fosas [77] or kinkajous (*Potos flavus*) [78]. There, results showed that while individuals ranged mostly solitarily, they nevertheless had an underlying complex social system, including spatial and/or temporal overlap [5,35,78], communal nesting [71], and associative or cooperative behaviour between individuals of one sex [5,35,74]. We suggest that the recognition of underlying 'hidden' complexities in these apparently 'solitary' organizations is long overdue and needs to be taken into account when investigating the causes and consequences of group living and social behaviour.

Ethics. The study was conducted under the permission of the ethical committee of Pretoria University and the Northern Cape Conservation Service, South Africa (permit number: EC054-10). Details have been included in the Methods section of the main text.

Data accessibility. The datasets supporting this article have been uploaded as part of the electronic supplementary material.

Authors' contributions. B.G. carried out the fieldwork, analysed the data, conducted the genetics analysis and wrote the manuscript. B.K. carried out the general linear mixed effect model, created figure 1, and revised the manuscript. M.B.M. supervised and financed this study and revised the manuscript. B.G. and M.B.M. collaborated in the conception and design of this study. All authors gave final approval for publication.

Competing interests. The authors have no competing interests.

Funding. Financial support for the study was provided by the University of Zurich.

Acknowledgements. We thank T.H. Clutton-Brock and the Kalahari Research Trust for letting us work and use the facilities at the Kuruman River Reserve. We are grateful to the families J. & S. Koetze, P. & M. Koetze and F. & L. de Bruin for allowing us to work on their lands. Invaluable help during fieldwork and data collection was provided by D. Jansen, K.-L. Roelofse, M. Tharavajah, T. Schellenberg, N. Milling, C. Young, N. Harrison, H. Stühlen, C. Sanderson, and C. Prussick. We are thankful to the managers of the meerkat project, T. Flower, R. Sutcliffe, D. Bell, M. Price, C. Borgeaud, J. Samson and N. Thavarajah, for the continued support during this study. A big thank you goes to all the meerkat volunteers and researchers that reported slender sightings. Beat Naef-Danzer provided essential information and support with the triangulation study. We thank Kamran Safi for his help with the spatial analysis and for providing R protocols for our analysis.

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
