## [Reviewer comments · Royal Society Open Science]

Review History

RSOS-182160.R0 (Original submission)

Review form: Reviewer 1 (Peter Kappeler)

Is the manuscript scientifically sound in its present form?

Yes

Are the interpretations and conclusions justified by the results?

Yes

Is the language acceptable?

Yes

Is it clear how to access all supporting data?

Yes

Do you have any ethical concerns with this paper?

No

Have you any concerns about statistical analyses in this paper?

No

Recommendation?

Accept with minor revision (please list in comments)

Comments to the Author(s)

Graw et al. Present novel and interesting data on the social organization of a little-studied mongoose species. Because mongooses, as a group, remain understudied, this report will make an important contribution to mammalian socioecology. In addition, the existence of male associations in slender mongooses has been reported only anecdotally. Because this aspect of their social organization is rare among mammals, the details provided by the present ms are most welcome for those interested in carnivore/mammalian sociality. The data set is impressive, and the analyses make good use of this novel information. Some points require clarification and/or re-organization, however, but they are mainly limited to the Introduction and Discussion, so that only relatively minor revisions are necessary to make this ms acceptable for publication.

On conceptual point requiring revision is evident from the first two sentences of the abstract already. The authors are interested in several aspects of the social system of this species, but exact definitions and relationships among the different components remain unclear. They also interpret their findings with respect to the level of social complexity, which is intimately related to the first aspect. These conceptual relationships have been defined and characterized in a recent review (Kappeler P.M. 2019 A framework for studying social complexity. *Behavioral Ecology and Sociobiology* 73, 13. (doi:10.1007/s00265-018-2601-8) that was not yet published when this ms was drafted. It may provide some structural and semantic guidance for framing their study aims and predictions. I am, of course, not insisting that the authors follow my classification scheme, but I think that it will help clarify their thinking and writing about social organization, social structure and aspects of the mating system.

L. 68: male alliances were also reported for the narrow-striped mongoose (Schneider TC, Kappeler PM (2016) Gregarious sexual segregation: the unusual social organization of the Malagasy narrow-striped mongoose (*Mungotictis decemlineata*) *Behavioral Ecology and Sociobiology* 70:913-926 doi:10.1007/s00265-016-2113-3).

The summary of mongoose social systems (line 79 ff) may perhaps benefit from another fairly recent review of this topic (Schneider TC, Kappeler PM (2014) Social systems and life-history characteristics of mongooses *Biological Reviews* 89:173-198 doi:10.1111/brv.12050).

The Introduction and Discussion could be better structured by inserting paragraph breaks whenever the focus moves on to another topic (e.g. line 84 before "Here, we investigate..."); this would enhance readability throughout the manuscript. In this context, I would also like to propose to add subheaders to the Discussion.

L 152: it is not obvious in the Abstract that radio-tracking was limited to resting animals in burrows. (In the Results, the authors speak of "sleeping site home ranges"). This detail should be mentioned there, and its implications/limitations should be discussed in more detail.

L 311: the wording a/o definition is a bit awkward here: "Spatial groups consisted of 1 to 3 adult males". I would not call a single individual a group. This variation may actually also warrant some additional analyses/comment: do single males differ from the groups in terms of body size, number of (male) relatives alive or some other relevant trait?

L 340 The reproductive success per male association is an unusual and presumably misleading measure because it does apparently not take variation in the number of males/members into account. Related to the previous inquiry, it would be interesting to present (average) male reproductive success for single males and for associations of 2 and 3 males separately.

L 477 “behaved mostly solitary” sounds awkward; “ranged” might work better as a verb here.

L 480: the “hidden complexities” appear as a prominent conclusion in the abstract, but this point is not elaborated/discussed in the Discussion; it only appears at the end of the Conclusions. Given the large number of still unstudied (presumably) solitary carnivores/mammals, it would be helpful to elaborate a bit on which hidden complexities were hidden for which reason, why/how they were brought to the foreground and what exactly the complexity part entails. Sorry to refer to another earlier study of mine (Kappeler PM, Wimmer B, Zinner D, Tautz D (2002) The hidden matrilineal structure of a solitary lemur: implications for primate social evolution. *Proceedings of the Royal Society of London B* 269:1755-1763 doi:10.1098/rspb.2002.2066), but it provides another example of this point that I find worth emphasizing.

I am really sorry for highlighting some of my own previous papers; I hate reviews where referees say “you should cite me here or there”, but it will hopefully be evident that these recommendations are motivated by an attempt to make an interesting and very good paper even a little bit better.

Peter Kappeler

Review form: Reviewer 2 (Dr Hazel Nichols)

Is the manuscript scientifically sound in its present form?

Yes

Are the interpretations and conclusions justified by the results?

Yes

Is the language acceptable?

Yes

Is it clear how to access all supporting data?

Yes

Do you have any ethical concerns with this paper?

No

Have you any concerns about statistical analyses in this paper?

No

Recommendation?

Accept with minor revision (please list in comments)

Comments to the Author(s)

See attached file (Appendix A).

Decision letter (RSOS-182160.R0)

27-Feb-2019

Dear Mrs Graw

On behalf of the Editors, I am pleased to inform you that your Manuscript RSOS-182160 entitled "Social organization of a solitary carnivore: spatial behaviour, interactions and relatedness in the slender mongoose" has been accepted for publication in Royal Society Open Science subject to minor revision in accordance with the referee suggestions. Please find the referees' comments at the end of this email.

The reviewers and handling editors have recommended publication, but also suggest some minor revisions to your manuscript. Therefore, I invite you to respond to the comments and revise your manuscript.

- Ethics statement

- Data accessibility

<http://datadryad.org/submit?journalID=RSOS&manu=RSOS-182160>

- Competing interests

- Authors' contributions

- Acknowledgements

- Funding statement

Because the schedule for publication is very tight, it is a condition of publication that you submit the revised version of your manuscript before 08-Mar-2019. Please note that the revision deadline will expire at 00.00am on this date. If you do not think you will be able to meet this date please let me know immediately.

- 1) A text file of the manuscript (tex, txt, rtf, docx or doc), references, tables (including captions) and figure captions. Do not upload a PDF as your "Main Document";
- 2) A separate electronic file of each figure (EPS or print-quality PDF preferred (either format should be produced directly from original creation package), or original software format);
- 3) Included a 100 word media summary of your paper when requested at submission. Please ensure you have entered correct contact details (email, institution and telephone) in your user account;
- 4) Included the raw data to support the claims made in your paper. You can either include your data as electronic supplementary material or upload to a repository and include the relevant doi

within your manuscript. Make sure it is clear in your data accessibility statement how the data can be accessed;

5) All supplementary materials accompanying an accepted article will be treated as in their final form. Note that the Royal Society will neither edit nor typeset supplementary material and it will be hosted as provided. Please ensure that the supplementary material includes the paper details where possible (authors, article title, journal name).

on behalf of Dr Alexander Ophir (Associate Editor) and Professor Kevin Padian (Subject Editor)
openscience@royalsociety.org

Associate Editor Comments to Author (Dr Alexander Ophir):

Dear Dr. Graw,

I have now received reviews of your manuscript from two experts in this field. As you can see from their comments they were both impressed by the study you described and the significant data set that accompanied it. Although both raised several points that you will need to address, all these comments seem to be relatively minor and easily addressed. Please give special thought to Reviewer 1's suggestion to use the recent Review by Kappler et al to provide additional structure and enhance the framework within which you are discussing your study. I think this will potentially even further broaden the appeal of your paper to a wider RSOS audience. These points notwithstanding, congratulations on a very nice paper.

Alex Ophir
Associate Editor

Reviewer comments to Author:

Reviewer: 1

Comments to the Author(s)

Graw et al. Present novel and interesting data on the social organization of a little-studied mongoose species. Because mongooses, as a group, remain understudied, this report will make an important contribution to mammalian socioecology. In addition, the existence of male associations in slender mongooses has been reported only anecdotally. Because this aspect of their social organization is rare among mammals, the details provided by the present ms are most welcome for those interested in carnivore/mammalian sociality. The data set is impressive, and the analyses make good use of this novel information. Some points require clarification and/or re-organization, however, but they are mainly limited to the Introduction and Discussion, so that only relatively minor revisions are necessary to make this ms acceptable for publication.

On conceptual point requiring revision is evident from the first two sentences of the abstract already. The authors are interested in several aspects of the social system of this species, but exact definitions and relationships among the different components remain unclear. They also interpret their findings with respect to the level of social complexity, which is intimately related to the first aspect. These conceptual relationships have been defined and characterized in a recent review (Kappeler P.M. 2019 A framework for studying social complexity. *Behavioral Ecology and Sociobiology* 73, 13. (doi:10.1007/s00265-018-2601-8) that was not yet published when this ms was drafted. It may provide some structural and semantic guidance for framing their study aims and predictions. I am, of course, not insisting that the authors follow my classification scheme, but I think that it will help clarify their thinking and writing about social organization, social structure and aspects of the mating system.

L. 68: male alliances were also reported for the narrow-striped mongoose (Schneider TC, Kappeler PM (2016) Gregarious sexual segregation: the unusual social organization of the Malagasy narrow-striped mongoose (*Mungotictis decemlineata*) *Behavioral Ecology and Sociobiology* 70:913-926 doi:10.1007/s00265-016-2113-3).

The summary of mongoose social systems (line 79 ff) may perhaps benefit from another fairly recent review of this topic (Schneider TC, Kappeler PM (2014) Social systems and life-history characteristics of mongooses *Biological Reviews* 89:173-198 doi:10.1111/brv.12050).

The Introduction and Discussion could be better structured by inserting paragraph breaks whenever the focus moves on to another topic (e.g. line 84 before "Here, we investigate..."); this would enhance readability throughout the manuscript. In this context, I would also like to propose to add subheaders to the Discussion.

L 152: it is not obvious in the Abstract that radio-tracking was limited to resting animals in burrows. (In the Results, the authors speak of "sleeping site home ranges"). This detail should be mentioned there, and its implications/limitations should be discussed in more detail.

L 311: the wording a/o definition is a bit awkward here: "Spatial groups consisted of 1 to 3 adult males". I would not call a single individual a group. This variation may actually also warrant some additional analyses/comment: do single males differ from the groups in terms of body size, number of (male) relatives alive or some other relevant trait?

L 340 The reproductive success per male association is an unusual and presumably misleading measure because it does apparently not take variation in the number of males/members into

account. Related to the previous inquiry, it would be interesting to present (average) male reproductive success for single males and for associations of 2 and 3 males separately.

L 477 "behaved mostly solitary" sounds awkward; "ranged" might work better as a verb here.

L 480: the "hidden complexities" appear as a prominent conclusion in the abstract, but this point is not elaborated/discussed in the Discussion; it only appears at the end of the Conclusions. Given the large number of still unstudied (presumably) solitary carnivores/mammals, it would be helpful to elaborate a bit on which hidden complexities were hidden for which reason, why/how they were brought to the foreground and what exactly the complexity part entails. Sorry to refer to another earlier study of mine (Kappeler PM, Wimmer B, Zinner D, Tautz D (2002) The hidden matrilineal structure of a solitary lemur: implications for primate social evolution. *Proceedings of the Royal Society of London B* 269:1755-1763 doi:10.1098/rspb.2002.2066), but it provides another example of this point that I find worth emphasizing.

I am really sorry for highlighting some of my own previous papers; I hate reviews where referees say "you should cite me here or there", but it will hopefully be evident that these recommendations are motivated by an attempt to make an interesting and very good paper even a little bit better.

Peter Kappeler

Reviewer: 2

Comments to the Author(s)
See attached file

Author's Response to Decision Letter for (RSOS-182160.R0)

See Appendix B.

Decision letter (RSOS-182160.R1)

02-Apr-2019

Dear Mrs Graw,

I am pleased to inform you that your manuscript entitled "Social organization of a solitary carnivore: spatial behaviour, interactions and relatedness in the slender mongoose" is now accepted for publication in Royal Society Open Science.

on behalf of Dr Alexander Ophir (Associate Editor) and Kevin Padian (Subject Editor)
openscience@royalsociety.org

Associate Editor Comments to Author (Dr Alexander Ophir):
Associate Editor: 1
Comments to the Author:
(There are no comments.)

Reviewer comments to Author:

Appendix A

This study combines genetic and behavioural data to further our understanding of the social and breeding system of the slender mongoose. I find it to be generally well-written, with the introduction nicely setting up the background of the study, and the discussion putting the results into context. As the authors point out, studies of mammalian species that are solitary and difficult to habituate are rare and hence the present study provides a valuable contribution to our understanding of mammalian social biology. I suggest some additional clarification in the methods and results, particularly clarifying ethical aspects, sample sizes and figure 3. However, on the whole, I think this paper is a valuable contribution to the literature.

Abstract:

L18-19: This sentence could be clearer. At the moment, it's not entirely clear what overlaps what. Consider splitting into two sentences, one focused on males and one on females.

Introduction: Generally very nicely written, with the background and context of the study well explained.

L37: typo? Should this be 'male alliance formation'?

L64-65: I think that the sentence would be clearer with 'especially in species with synchronised estrus' in brackets.

L83: It would make sense to include a reference for the long-term study of dwarf mongooses, since you include references for the other two studies/species.

L95: It's not very clear what 'the other studies' means. Do you mean all other published work on this species? Perhaps rephrase to 'Previous studies ...'

Methods: Generally clear, but ethical information is missing. Could you include details of the weight of collars as a proportion of the body weight, details on how stress was minimised during capture. How long did captures take? How frequently were traps checked? Was any food/water provided? Were individuals monitored after anaesthesia? Were there any adverse effects?

It would also be good if you could include an explanation of exactly how you defined spatial groups here. I know you include a general definition in the introduction, but one more specific to your study would help to justify your categorisations.

L141: 'During captures' is a little confusing. Perhaps re-phrase as 'Over our study period, we made a total of 215 captures, comprising 131 unique individuals...'

L144-145: 'Whenever possible we radio-collared at least two adults in close proximity.' is not very clear. Could you clarify? Details of the number of individuals collared would be useful here. I assume you didn't collar all 131 individuals? Given that tracking details are in the following paragraph, perhaps move this sentence to the next paragraph.

L127 (Capture procedure): No details of anaesthetic are given. I assume you anaesthetised animals before ageing, sexing, microchipping and DNA sampling. Also, how was the tissue sample taken? E.g.

using surgical scissors/scalpel? How were infection chances minimised? Were the scissors/scalpel sterilised? Was any antiseptic applied?

L165-166: Are there any references for this software?

L171: remove comma.

L180-182: This sentence is confusing – comma use issue?

L194: What measure of relatedness did you use, and why? There are many slightly different measures with different benefits.

L201: 'less' should be 'fewer'

Results: Some of your results could be clearer. In particular, sample sizes need clarifying, figure legends could do with further explanation, and figure 3 is particularly confusing and could be replaced with something more intuitive.

L212: The number of individuals you've included is lower than the number you report in the methods. I assume this is because you didn't get enough data from some individuals, but it'd be good to clarify this.

L266: What are the 'original' animals?

Figure 1: Three of the five groups seem to consist of just one male and one female. This doesn't come across in the description of the results. From looking at Figure 3, there seem to be more individuals present in your core study population than appear in Figure 1. Have you excluded individuals born during the study period (or some other such rule)? It should be made very clear what your criteria for inclusion in this analysis was and what the sample sizes are for each analysis.

L237-239: This sentence is a little confusing. Without the brackets, it reads 'male ranges remaining to 83 % and females to 85.5 % the same'. It is not clear exactly what 'the same' means in this context. Do you mean that the ranges of individuals had on average 83% (males) and 85.5% (females) overlap with the range of the same individual in the previous year?

L245: 'actual' is not necessary, here or in other places e.g. L253.

L245: 'Alternate' is the incorrect word (it means to occur in turn, or every other). I think you mean 'Alternative'.

L253: It'd be good to compare the range shifts with the total size of the ranges. E.g. what are the minimum and maximum dimensions of ranges?

L256-259: It is a little confusing here which figures are from the breeding vs non-breeding season. Could you clarify which figures are for which in the brackets? Did they use more burrows in the breeding season?

L265-269: this seems a little counter intuitive on first reading. If male ranges increase in size during the breeding season, and overlap more with other spatial groups in the breeding season, then wouldn't we expect greater overlap between males and females during the breeding season? Is this

because male ranges are expanding into areas without females? This paragraph seems a bit confusing over-all. Can you think of a way to clarify it so it's clear what is increasing and decreasing and why (i.e. how these increases and decreases relate to each other)?

L270: both of the figures in brackets are for 'breeding'. I think one should be 'non-breeding'.

L288: Please include full model details somewhere, including effect sizes.

L292: You mention significant differences (or lack thereof) but do not present test statistics.

L301: Appropriate caution should be used to interpret individuals being within 100m of each other to be socialising or foraging together. Could two individuals not be within 100m and unaware of each other? Perhaps behavioural observations you made during your study suggest that individuals within 100m are always aware of each other? If so, it would strengthen your argument to include further justification for assuming that individuals within 100m are aware of each other or are socialising.

L310: How many relatedness values (and individuals) are you using for this analysis? Is it limited to those individuals in your core study area (Fig 1), or the wider population? If the wider population, then how did you define social groupings?

L312: Should this be 'A Kruskal-Wallis....' There are a few grammatical errors/typos in this sentence/paragraph so please check through.

Figure 3: I find this figure very confusing! There must be a clearer way of presenting it. Something like Figure 4 in Griffin et al (2003) would be good if it's possible to create this for your small sample sizes. Perhaps also include your sample sizes in the figure too (e.g. in brackets next to the appropriate mean relatedness value).

Griffin, A.S., Pemberton, J.M., Brotherton, P.N., McIlrath, G., Gaynor, D., Kansky, R., O'riain, J. and Clutton-Brock, T.H., 2003. A genetic analysis of breeding success in the cooperative meerkat (*Suricata suricatta*). *Behavioral Ecology*, 14(4), pp.472-480.'

L340: Could you clarify what you mean by 'associations' here. Do you mean that males within associations of close kin fathered more pups per capita than males that were not in associations with kin? Also, what kind of average are you using here? Values seem to be much higher than the median value you presented earlier. Are you using means instead? If so, why? It would also be good to include sample sizes where you quote your statistics. As you mention, sample sizes are small, so it'd be good to include them so the reader can interpret with appropriate caution.

L350: For what proportion of litters did you detect multiple paternity? Note that this proportion should be out of the total number of litters where it was possible to detect multiple paternity (i.e. where more than one offspring had been assigned parentage). This might give a feeling for how common multiple paternity is.

Table 3: Could do with further clarification. From what I can tell, you present the total number of males present within kin associations and without kin associations, but then present the average (mean I assume?) of females and pups. Would it make more sense to present the average number of males instead of (or as well as) the total, so we can get an idea of what a 'typical' kin association

group and a 'typical' non-kin group looks like? Also, could you clarify how these averages were calculated? E.g. are you presenting the average number of pups a male produces per year, or the average production over the 3 years of your study? Including ranges or another measure of variance may also be useful.

Discussion:

L363: it is unclear whether 'their' offspring refers to the offspring of the males or females.

L366: it might be worth reminding the reader that sample sizes are small so results should be interpreted with caution and further studies are required to fully understand the intricacies of the social system of this species.

L373: perhaps add a bit of caution here as you've not directly tested this. Eg. 'This pattern most likely reflects sexual differences in resource competition'.

L389-391: I find this last sentence a bit unclear.

L392: But you don't actually give your definition in the methods.

L412: should this be 'studies' as you cite two.

L430: should this be 'slender mongoose pups'.

L448: issue with comma use. Should one be a full stop or colon?

L460-462: It's not entirely clear which species you're referring to in this last sentence.

Supplementary material: As with the main paper, there are elements that could do with additional explanation e.g. for Table B it is not clear whether the numbers represent range sizes or overlap data. For example are values in the column 2008-2009 overlap values between 2008 and 2009, or is it the size of the territory in the 2008-2009 field season. Also, what do 'overlap 1' and 'overlap 2' represent?

Appendix B

Reviewer comments to Author:

Reviewer: 1

Comments to the Author(s)

Graw et al. Present novel and interesting data on the social organization of a little-studied mongoose species. Because mongooses, as a group, remain understudied, this report will make an important contribution to mammalian socioecology. In addition, the existence of male associations in slender mongooses has been reported only anecdotally. Because this aspect of their social organization is rare among mammals, the details provided by the present ms are most welcome for those interested in carnivore/mammalian sociality. The data set is impressive, and the analyses make good use of this novel information. Some points require clarification and/or re-organization, however, but they are mainly limited to the Introduction and Discussion, so that only relatively minor revisions are necessary to make this ms acceptable for publication.

On conceptual point requiring revision is evident from the first two sentences of the abstract already. The authors are interested in several aspects of the social system of this species, but exact definitions and relationships among the different components remain unclear. They also interpret their findings with respect to the level of social complexity, which is intimately related to the first aspect. These conceptual relationships have been defined and characterized in a recent review (Kappeler P.M. 2019 A framework for studying social complexity. Behavioral Ecology and Sociobiology 73, 13. (doi:10.1007/s00265-018-2601-8) that was not yet published when this ms was drafted. It may provide some structural and semantic guidance for framing their study aims and predictions. I am, of course, not insisting that the authors follow my classification scheme, but I think that it will help clarify their thinking and writing about social organization, social structure and aspects of the mating system.

- Thank you, that was a very helpful paper. We included some of your suggestions into our introduction (L 41-45).

L. 68: male alliances were also reported for the narrow-striped mongoose (Schneider TC, Kappeler PM (2016) Gregarious sexual segregation: the unusual social organization of the Malagasy narrow-striped mongoose (*Mungotictis decemlineata*) Behavioral Ecology and Sociobiology 70:913-926 doi:10.1007/s00265-016-2113-3).

- Thank you for pointing this out. As you explain in your paper together with T. Schneider (2016), male and female narrow-striped mongooses form associations as a predator-avoidance strategy, in fact you state that males do give up these associations during the mating season in the pursuit of receptive females. Male alliances as we mention them here, are associations of males in order to gain reproductive benefits and access to otherwise less defensible females. We therefore suggest, and you mention this in the discussion of your paper as well, that the narrow-striped mongoose is not an example of a species forming such male (reproductive) alliances.

The summary of mongoose social systems (line 79 ff) may perhaps benefit from another fairly recent review of this topic (Schneider TC, Kappeler PM (2014) Social systems and life-history characteristics of mongooses Biological Reviews 89:173-198 doi:10.1111/brv.12050).

- We agree and have included this reference (L 84-89).

The Introduction and Discussion could be better structured by inserting paragraph breaks whenever the focus moves on to another topic (e.g. line 84 before “Here, we investigate...”); this would enhance readability throughout the manuscript. In this context, I would also like to propose to add subheaders to the Discussion.

- We have included several paragraph breaks, including the one mentioned by you in line 84. We have also added subheaders to the discussion.

L 152: it is not obvious in the Abstract that radio-tracking was limited to resting animals in burrows. (In the Results, the authors speak of “sleeping site home ranges”). This detail should be mentioned there, and its implications/limitations should be discussed in more detail.

- As the abstract is limited to 200 words it is unfortunately not possible to go into details about the tracking routine there, but we mention now that we calculated sleeping ranges (L 19). We explained this in more detail in the methods (L 156-160) and also discussed possible implications in the discussion part (L 539-544).

L 311: the wording a/o definition is a bit awkward here: “Spatial groups consisted of 1 to 3 adult males”. I would not call a single individual a group. This variation may actually also warrant some additional analyses/comment: do single males differ from the groups in terms of body size, number of (male) relatives alive or some other relevant trait?

- Spatial groups consisted of 1-3 adults males AND 1-4 adult females. Thank you for pointing out, that our wording is not clear enough here. We did look into body size differences between associated and non-associated males and could find none, same goes for any other obvious trait. What we did find and this is mentioned in the manuscript is a slight variances in reproductive success (L 358-359).

L 340 The reproductive success per male association is an unusual and presumably misleading measure because it does apparently not take variation in the number of males/members into account. Related to the previous inquiry, it would be interesting to present (average) male reproductive success for single males and for associations of 2 and 3 males separately.

- We agree and have therefore added information about the number of pups sired by an associated vs a non-associated male (L 393-395). Unfortunately sample sizes were too small to compare males in 2 vs 3 male associations.

L 477 “behaved mostly solitary” sounds awkward; “ranged” might work better as a verb here.

- changed as suggested (L 555)

L 480: the “hidden complexities” appear as a prominent conclusion in the abstract, but this point is not elaborated/discussed in the Discussion; it only appears at the end of the Conclusions. Given the large number of still unstudied (presumably) solitary carnivores/mammals, it would be helpful to elaborate a bit on which hidden complexities were hidden for which reason, why/how they were brought to the foreground and what exactly the complexity part entails. Sorry to refer to another earlier study of mine (Kappeler PM, Wimmer B, Zinner D, Tautz D (2002) The hidden

matrilineal structure of a solitary lemur: implications for primate social evolution. Proceedings of the Royal Society of London B 269:1755-1763 doi:10.1098/rspb.2002.2066), but it provides another example of this point that I find worth emphasizing.

- Thank you for pointing this out, we added a paragraph addressing social complexity in the slender mongoose to the manuscript and included your reference (L 527-536).

I am really sorry for highlighting some of my own previous papers; I hate reviews where referees say “you should cite me here or there”, but it will hopefully be evident that these recommendations are motivated by an attempt to make an interesting and very good paper even a little bit better.

- Many thanks for the very helpful comments and we should have brought in some of your papers in the first round. It certainly improves the work.

Peter Kappeler

Reviewer 2:

This study combines genetic and behavioural data to further our understanding of the social and breeding system of the slender mongoose. I find it to be generally well-written, with the introduction nicely setting up the background of the study, and the discussion putting the results into context. As the authors point out, studies of mammalian species that are solitary and difficult to habituate are rare and hence the present study provides a valuable contribution to our understanding of mammalian social biology. I suggest some additional clarification in the methods and results, particularly clarifying ethical aspects, sample sizes and figure 3. However, on the whole, I think this paper is a valuable contribution to the literature.

Abstract:

L18-19: This sentence could be clearer. At the moment, it's not entirely clear what overlaps what. Consider splitting into two sentences, one focused on males and one on females.

- We rearranged this sentence and split it in two (L 17-20).

Introduction: Generally very nicely written, with the background and context of the study well explained.

L37: typo? Should this be ‘male alliance formation’

- Yes, thank you for pointing this out. Has been changed accordingly (L 37).

L64-65: I think that the sentence would be clearer with 'especially in species with synchronised estrus' in brackets.

- changed accordingly (L 69-70)

L83: It would make sense to include a reference for the long-term study of dwarf mongooses, since you include references for the other two studies/species.

- Thank you for pointing this out, we agree and have included a reference for the dwarf mongooses (L 88).

L95: It's not very clear what 'the other studies' means. Do you mean all other published work on this species? Perhaps rephrase to 'Previous studies ...'

- changed to 'previous studies' (L 100)

Methods: Generally clear, but ethical information is missing. Could you include details of the weight of collars as a proportion of the body weight, details on how stress was minimised during capture. How long did captures take? How frequently were traps checked? Was any food/water provided? Were individuals monitored after anaesthesia? Were there any adverse effects?

- We included a detailed ethical statement answering those questions (L 218-250). If this is too elaborate, we can shorten according to any feedback.

It would also be good if you could include an explanation of exactly how you defined spatial groups here. I know you include a general definition in the introduction, but one more specific to your study would help to justify your categorisations.

- We added some information on how we determined spatial groups into the methods section (L 172-175).

L141: 'During captures' is a little confusing. Perhaps re-phrase as 'Over our study period, we made a total of 215 captures, comprising 131 unique individuals...'

- We rephrased this sentence making it clearer (L 146-147).

L144-145: 'Whenever possible we radio-collared at least two adults in close proximity.' is not very clear. Could you clarify? Details of the number of individuals collared would be useful here. I assume you didn't collar all 131 individuals? Given that tracking details are in the following paragraph, perhaps move this sentence to the next paragraph.

- We moved this sentence into the next paragraph and added 'caught in close proximity',

hopefully making this more clear (L 153).

L127 (Capture procedure): No details of anaesthetic are given. I assume you anaesthetised animals before ageing, sexing, microchipping and DNA sampling. Also, how was the tissue sample taken? E.g. using surgical scissors/scalpel? How were infection chances minimised? Were the scissors/scalpel sterilised? Was any antiseptic applied?

- We have included details on this procedure in the ethical statement (L 218-250).

L165-166: Are there any references for this software?

- We have added the reference (L 172).

L171: remove comma.

- Done (L 175).

L180-182: This sentence is confusing – comma use issue?

- We added to this sentence in order to make it more clear that animals had to be located from both receiver stations within a two minute time frame (L 191-192).

L194: What measure of relatedness did you use, and why? There are many slightly different measures with different benefits.

- We used the Queller & Goodnight estimator. We specifically mention this now (L 205-206).

L201: 'less' should be 'fewer'

- changed accordingly (L 211)

Results: Some of your results could be clearer. In particular, sample sizes need clarifying, figure legends could do with further explanation, and figure 3 is particularly confusing and could be replaced with something more intuitive.

L212: The number of individuals you've included is lower than the number you report in the methods. I assume this is because you didn't get enough data from some individuals, but it'd be good to clarify this.

- We mention that we calculated ranges for the 26 slender mongooses 'that were tracked at least 50 times during the study period (range: 51-412 trackings, mean \pm SE: 126 trackings \pm 25) and 30 times annually'. (L 255-258)

L266: What are the 'original' animals?

- We meant to say the 'animals originally caught in 2008' at the beginning of our study and have changed this accordingly (L 271-272).

Figure 1: Three of the five groups seem to consist of just one male and one female. This doesn't come across in the description of the results. From looking at Figure 3, there seem to be more individuals present in your core study population than appear in Figure 1. Have you excluded individuals born during the study period (or some other such rule)? It should be made very clear what your criteria for inclusion in this analysis was and what the sample sizes are for each analysis.

- Figure 1 only shows the animals that were originally collared at the beginning of the study in 2008. This figure is supposed to give an overview of range sizes in the different sexes and overlap patterns within and between spatial groups. Consecutively more individuals were collared and in some cases these original animals died or disappeared and were replaced by new ones, therefore the difference in sample size. Displaying all collared animals would have made the figure too crowded, also not all tracked animals were alive at the same time. We clarified this in the figure legend (L 270-271).

L237-239: This sentence is a little confusing. Without the brackets, it reads 'male ranges remaining to 83 % and females to 85.5 % the same'. It is not clear exactly what 'the same' means in this context. Do you mean that the ranges of individuals had on average 83% (males) and 85.5% (females) overlap with the range of the same individual in the previous year?

- Yes, as indicated in the methods section (L 176), we compared the overlap of ranges of individuals in consecutive years, ranges overlapped around 83% / 85.5% with those of previous years. We changed this sentence to hopefully make it more clear (L 282-285).

L245: 'actual' is not necessary, here or in other places e.g. L253.

- We deleted these.

L245: 'Alternate' is the incorrect word (it means to occur in turn, or every other). I think you mean 'Alternative'.

- Thank you for seeing this, changed accordingly (L 290).

L253: It'd be good to compare the range shifts with the total size of the ranges. E.g. what are the minimum and maximum dimensions of ranges?

- Mean annual and total range sizes are given above (L 258-261). For males minimum

total range size was 1 km², maximum was 6.21 km²; for females total ranges varied between 0.53 km² and 2.06 km², we added this information into the manuscript (L 298-299).

L256-259: It is a little confusing here which figures are from the breeding vs non-breeding season. Could you clarify which figures are for which in the brackets? Did they use more burrows in the breeding season?

- Yes, they do use more burrows during the breeding season. We changed the order and wording in the brackets and hopefully made it more clear (L 304-305).

L265-269: this seems a little counter intuitive on first reading. If male ranges increase in size during the breeding season, and overlap more with other spatial groups in the breeding season, then wouldn't we expect greater overlap between males and females during the breeding season? Is this because male ranges are expanding into areas without females? This paragraph seems a bit confusing over-all. Can you think of a way to clarify it so it's clear what is increasing and decreasing and why (i.e. how these increases and decreases relate to each other)?

- We rephrased this (L 311-314). Because male ranges are bigger during the breeding season, less percentage of male ranges gets overlapped by female ranges during the breeding season.

L270: both of the figures in brackets are for 'breeding'. I think one should be 'non-breeding'.

- You are correct, we corrected this (L 316).

L288: Please include full model details somewhere, including effect sizes.

- We have included effect sizes and confidence intervals in the text as well as the spread of the random effects (L 331-335).

L292: You mention significant differences (or lack thereof) but do not present test statistics.

- We added statistics here (L 341).

L301: Appropriate caution should be used to interpret individuals being within 100m of each other to be socialising or foraging together. Could two individuals not be within 100m and unaware of each other? Perhaps behavioural observations you made during your study suggest that individuals within 100m are always aware of each other? If so, it would strengthen your argument to include further justification for assuming that individuals within 100m are aware of each other or are socialising.

- We agree, it is possible that mongooses at 100m are still unaware of each other and we cannot exclude that. We rephrased this sentence to explain that we do not mean to imply that they were always aware of each other (L 348).

L310: How many relatedness values (and individuals) are you using for this analysis? Is it limited to those individuals in your core study area (Fig 1), or the wider population? If the wider population, then how did you define social groupings?

- As mentioned in the methods part, we used adult individuals from 8 different spatial groups throughout the study area, a total of 68 individuals. We added this information in the results section (L 360). Spatial groups were defined by spatial behaviour (see results section on home range size) and social behaviour/communal denning.

L312: Should this be 'A Kruskal-Wallis....' There are a few grammatical errors/typos in this sentence/paragraph so please check through.

- Thank you for pointing this out, we corrected this and made changes throughout this paragraph (L 359-367).

Figure 3: I find this figure very confusing! There must be a clearer way of presenting it. Something like Figure 4 in Griffin et al (2003) would be good if it's possible to create this for your small sample sizes. Perhaps also include your sample sizes in the figure too (e.g. in brackets next to the appropriate mean relatedness value).

Griffin, A.S., Pemberton, J.M., Brotherton, P.N., McIlrath, G., Gaynor, D., Kinsky, R., O'riain, J. and Clutton-Brock, T.H., 2003. A genetic analysis of breeding success in the cooperative meerkat (*Suricata suricatta*). *Behavioral Ecology*, 14(4), pp.472-480.'

- The figure in Griffin et al. (2003) indicates pairwise relatedness values between individuals, while our figure depicts relatedness based on the parentage analysis. We tried to improve the figure by depicting males in blue letters and females in red, instead of indicating changes within spatial group composition with circles, we marked animals that died during the study period with a cross. We also added to the figure description (L 376-379).

L340: Could you clarify what you mean by 'associations' here. Do you mean that males within associations of close kin fathered more pups per capita than males that were not in associations with kin? Also, what kind of average are you using here? Values seem to be much higher than the median value you presented earlier. Are you using means instead? If so, why? It would also be good to include sample sizes where you quote your statistics. As you mention, sample sizes are small, so it'd be good to include them so the reader can interpret with appropriate caution.

- We added information to this paragraph, while before we only calculated average number of pups fathered by males within a spatial group vs unassociated males. We also added information about the rate of unsuccessful males (L 393-398).

L350: For what proportion of litters did you detect multiple paternity? Note that this proportion should be out of the total number of litters where it was possible to detect multiple paternity (i.e. where more than one offspring had been assigned parentage). This might give a feeling for how common multiple paternity is.

- Thank you for pointing this out. We successfully caught 4 complete litters with more than one pup (out of a total of 15 litters that emerged), of these 2 showed multiple paternities (50%). As sample sizes are so small, we do not feel that we can get a strong indication from this on how common multiple paternities are but have included this information in the manuscript now (L 406-408).

Table 3: Could do with further clarification. From what I can tell, you present the total number of males present within kin associations and without kin associations, but then present the average (mean I assume?) of females and pups. Would it make more sense to present the average number of males instead of (or as well as) the total, so we can get an idea of what a 'typical' kin association group and a 'typical' non-kin group looks like? Also, could you clarify how these averages were calculated? E.g. are you presenting the average number of pups a male produces per year, or the average production over the 3 years of your study? Including ranges or another measure of variance may also be useful.

- We changed table 3, so that it now shows individual reproductive success for all males captured and analysed throughout our study, rather than focussing on our main study groups and the success by associations. We changed the table description accordingly (L 410-411).

Discussion:

L363: it is unclear whether 'their' offspring refers to the offspring of the males or females.

- changed to 'independent' (L 422).

L366: it might be worth reminding the reader that sample sizes are small so results should be interpreted with caution and further studies are required to fully understand the intricacies of the social system of this species.

- We added that this is based on anecdotal data (L 424).

L373: perhaps add a bit of caution here as you've not directly tested this. Eg. 'This

pattern most likely reflects sexual differences in resource competition’.

- changed accordingly (L 434)

L389-391: I find this last sentence a bit unclear.

- We tried to make this sentence more clear, see L 451.

L392: But you don’t actually give your definition in the methods.

- We added the definition now (L 172-175).

L412: should this be ‘studies’ as you cite two.

- changed (L 475)

L430: should this be ‘slender mongoose pups’.

- changed (L 493-494)

L448: issue with comma use. Should one be a full stop or colon?

- changed (L 512)

L460-462: It’s not entirely clear which species you’re referring to in this last sentence.

- We added ‘in the white-tailed mongoose’ (L 525).

Supplementary material: As with the main paper, there are elements that could do with additional explanation e.g. for Table B it is not clear whether the numbers represent range sizes or overlap data. For example are values in the column 2008-2009 overlap values between 2008 and 2009, or is it the size of the territory in the 2008-2009 field season. Also, what do ‘overlap 1’ and ‘overlap 2’ represent?

- We added additional information (see supplementary material_tabel B).